elidelife

# The primary σ factor in *Escherichia coli* can access the transcription elongation complex from solution in vivo

Seth R Goldman[1,2†], Nikhil U Nair[1†], Christopher D Wells[1], Bryce E Nickels[2*], Ann Hochschild[1*]

[1]Department of Microbiology and Immunobiology, Harvard Medical School, Boston, United States; [2]Department of Genetics, Waksman Institute, Rutgers University, New Brunswick, United States

**Abstract** The σ subunit of bacterial RNA polymerase (RNAP) confers on the enzyme the ability to initiate promoter-specific transcription. Although σ factors are generally classified as initiation factors, σ can also remain associated with, and modulate the behavior of, RNAP during elongation. Here we establish that the primary σ factor in *Escherichia coli*, $\sigma^{70}$, can function as an elongation factor in vivo by loading directly onto the transcription elongation complex (TEC) *in trans*. We demonstrate that $\sigma^{70}$ can bind *in trans* to TECs that emanate from either a $\sigma^{70}$-dependent promoter or a promoter that is controlled by an alternative σ factor. We further demonstrate that binding of $\sigma^{70}$ to the TEC *in trans* can have a particularly large impact on the dynamics of transcription elongation during stationary phase. Our findings establish a mechanism whereby the primary σ factor can exert direct effects on the composition of the entire transcriptome, not just that portion that is produced under the control of $\sigma^{70}$-dependent promoters.

*For correspondence: bnickels@waksman.rutgers.edu (BEN); ahochschild@hms.harvard.edu (AH)

†These authors contributed equally to this work

Competing interests: The authors declare that no competing interests exist.

## Introduction

The σ subunit of bacterial RNA polymerase (RNAP) is an essential initiation factor that specifies the recognition of promoter sequences in the context of the RNAP holoenzyme (*Feklistov et al., 2014*). All bacteria contain a primary σ factor that directs transcription from the major class of bacterial promoters; in addition, most bacteria contain multiple alternative σ factors that direct transcription from specialized promoters in response to stress or alterations in growth state (*Gruber and Gross, 2003*; *Osterberg et al., 2011*; *Guo and Gross, 2014*). Among the best-studied primary σ factors is *Escherichia coli* $\sigma^{70}$, which recognizes promoters that are defined by two conserved hexameric DNA sequence elements termed the −10 and the −35 elements (consensus sequences: TATAAT and TTGACA, respectively). Members of the $\sigma^{70}$ family share a conserved 4-domain architecture, with domain 2 contacting the −10 element and domain 4 contacting the −35 element (*Gross et al., 1998*; *Paget and Helmann, 2003*; *Feklistov et al., 2014*; *Paget, 2015*). *E. coli* also has six alternative σ factors, five of which are members of the $\sigma^{70}$ family and recognize similarly positioned promoter elements using the counterparts of $\sigma^{70}$ domains 2 and 4. Most alternative σ factors exhibit highly restricted promoter specificity (*Koo et al., 2009b*; *Rhodius et al., 2013*). Thus, genes that are responsive to disparate physiological inputs often carry two or more promoters that are recognized by distinct σ factors (*Wade et al., 2006*; *Gama-Castro et al., 2008*; *Cho et al., 2014*).

Although σ factors were historically identified as promoter specificity factors, it has become clear that their roles are not limited to the initiation phase of transcription. In particular, multiple studies have shown that the release of σ from the transcription complex is not required for entry into the elongation phase of transcription (reviewed in *Mooney et al., 2005*; *Perdue and Roberts, 2011*).

**eLife digest** Proteins are made following instructions that are encoded by sections of DNA called genes. In the first step of protein production, an enzyme called RNA polymerase uses the gene as a template to make molecules of messenger ribonucleic acid (mRNA). This process—known as transcription—starts when RNA polymerase binds to a site at the start of a gene. The enzyme then moves along the DNA, assembling the mRNA as it goes. This stage of transcription is known as elongation and continues until the RNA polymerase reaches the end of the gene.

In bacteria, RNA polymerase needs a family of proteins called sigma factors to help it identify and bind to the start sites associated with the genes that will be transcribed. In the well studied bacterium known as *E. coli*, the primary sigma factor that is required for transcription initiation on most genes is called sigma 70. Recent research has shown that sigma 70 also influences the activity of RNA polymerase during elongation. During this stage, the RNA polymerase and several other proteins interact to form a complex called the transcription elongation complex (or TEC for short). However, it is not clear how sigma 70 gains access to this complex: does it simply remain with RNA polymerase after transcription starts, or is it freshly incorporated into the TEC during elongation?

Goldman, Nair et al. found that sigma 70 is able to incorporate into TECs during elongation and causes them to pause at specific sites in the gene. Sigma 70 can even incorporate into TECs on genes where transcription was initiated by a different sigma factor. These findings indicate that sigma 70 can directly influence the transcription of all genes, not just the genes with start sites that are recognized by this sigma factor.

Goldman et al. also observed that in cells that were growing and dividing rapidly, the pauses that occurred due to sigma 70 associating with TECs were of shorter duration than those in cells that were growing slowly. This implies that the growth status of the cells modulates the pausing of RNA polymerase during transcription. In the future, it will be important to understand how much influence the primary sigma factor has on RNA polymerase during elongation in *E. coli* and other bacteria.

Furthermore, the functional properties of a transcription elongation complex (TEC) containing σ differ from the properties of a TEC that does not contain σ. For example, TEC-associated $\sigma^{70}$ can induce transcription pausing by engaging promoter −10-like sequence elements within transcribed regions (*Ring et al., 1996*; *Brodolin et al., 2004*; *Nickels et al., 2004*; *Hatoum and Roberts, 2008*; *Deighan et al., 2011*; *Perdue and Roberts, 2011*), a phenomenon that was first uncovered in the context of the bacteriophage λ late gene promoter (reviewed in *Roberts et al., 1998*; *Perdue and Roberts, 2011*). This pausing occurs due to an interaction between the −10-like element and domain 2 of TEC-associated $\sigma^{70}$ (the same domain of $\sigma^{70}$ that binds the promoter −10 element during transcription initiation). In addition, the presence or absence of σ can alter the accessibility of the TEC to elongation factors, including the λ Q protein and RfaH (*Roberts et al., 1998*; *Nickels et al., 2002*, *2006*; *Sevostyanova et al., 2008*), and can influence the ability of RNAP to reinitiate transcription at certain promoters (*Bar-Nahum and Nudler, 2001*).

Initial-transcribed-region −10-like elements, such as those associated with the λ late promoters and the late promoters of other lambdoid phages, induce early elongation pausing because they are recognized by TECs that have not yet released the $\sigma^{70}$ that was used during initiation (*Marr et al., 2001*; *Mukhopadhyay et al., 2001*; *Nickels et al., 2004*; *Kapanidis et al., 2005*). In prior work, we showed that such promoter-proximal $\sigma^{70}$-dependent pause elements also function to inhibit $\sigma^{70}$ loss during the earliest stages of elongation, increasing the $\sigma^{70}$ content of downstream TECs (*Deighan et al., 2011*). This effect can be detected using a template that carries a second pause element positioned downstream of a promoter-proximal pause element; specifically, the presence of the promoter-proximal pause element facilitates the retention of $\sigma^{70}$ in the TEC and thus substantially enhances the extent of pausing induced by the downstream pause element both in vitro and in vivo (*Deighan et al., 2011*).

Although promoter −10-like elements that induce transcription pausing can be recognized by a $\sigma^{70}$ subunit that has been retained in the TEC after promoter escape, in vitro studies have established that transcribed region −10-like elements can also be recognized by a $\sigma^{70}$ subunit that was not present during initiation, but rather joined the TEC by loading *in trans* during elongation. Thus, it has been

shown that the efficiency of pausing induced by transcribed region −10-like elements can be increased in vitro by increasing the concentration of free σ$^{70}$ in the transcription reactions (*Mooney and Landick, 2003*; *Sevostyanova et al., 2008*; *Deighan et al., 2011*; *Sevostyanova et al., 2011*). A key question that emerges from these in vitro findings is whether or not cellular conditions permit σ$^{70}$ to gain access to the TEC through this 'trans-acting pathway' in vivo. Here we address this question by employing an assay that enables us to measure the extent of TEC pausing induced by a −10-like element within a transcribed region in vivo. We find that the extent of pausing induced by a transcribed-region −10-like element is sensitive to the intracellular concentration of σ$^{70}$, indicating that σ$^{70}$ can gain access to the TEC *in trans*. We further establish that σ$^{70}$ can gain access to the TEC *in trans* and engage −10-like elements within transcribed regions that are expressed under the control of either a σ$^{70}$-dependent promoter or a promoter that is recognized by an alternative σ factor. In addition, we show that the extent of TEC pausing mediated by σ$^{70}$ *trans* loading varies as a function of growth-phase. Our findings imply that distinct σ factors can control initiation and elongation on the same transcription unit in vivo, and that the functional consequences of σ$^{70}$ *trans* loading vary as a function of growth state.

## Results

### Detection of σ$^{70}$ *trans* loading on a σ$^{70}$-dependent transcription unit in vivo

To determine whether or not σ$^{70}$ can bind *in trans* to the TEC in vivo, we took advantage of the fact that TEC-associated σ$^{70}$ can induce transcription pausing by engaging transcribed-region −10-like elements. We therefore sought to determine whether or not the efficiency of pausing induced by a transcribed-region −10-like element was sensitive to the concentration of σ$^{70}$ present in vivo. To do this, we introduced into *E. coli* cells a plasmid carrying a σ$^{70}$-dependent promoter, λP$_{R'}$, fused to a transcribed region containing a −10-like element that has the potential to induce σ$^{70}$-dependent pausing at a nascent RNA length of ∼35 nt (*Deighan et al., 2011*) (*Figure 1A*, top); the transcription unit also contains an intrinsic terminator element (positioned to terminate transcription after the synthesis of an ∼116 nt transcript). Pausing induced by the −10-like element on this template in vitro is sensitive to the concentration of free σ$^{70}$ in the transcription reactions (*Deighan et al., 2011*); furthermore, because the template lacks a promoter-proximal −10-like element, engagement of the pause element by σ$^{70}$ that is retained during the transition from initiation to elongation contributes minimally to the observed pausing (*Deighan et al., 2011*).

We tested whether or not the efficiency of pausing at a nascent RNA length of ∼35 nt on this template was sensitive to the concentration of σ$^{70}$ present in vivo by introducing into the cells a second plasmid that did or did not direct the production of excess σ$^{70}$. To detect nascent RNAs associated with paused TECs (pause RNAs) and full-length terminated transcripts produced from this template, we isolated total RNA and used Northern blotting with a locked-nucleic-acid (LNA) probe, as described previously (*Deighan et al., 2011*). We quantified the extent of pausing by dividing the signal associated with a pause RNA by the sum of this signal and the signal associated with the full-length terminated transcript (hereafter termed relative abundance). We found that the relative abundance of a ∼35-nt pause RNA (see *Deighan et al., 2011*) increased ∼fivefold when σ$^{70}$ was overproduced by a factor of ∼7, compared to that observed in cells containing chromosomally encoded σ$^{70}$ only (*Figure 1A*). Furthermore, the ∼35-nt pause RNA was barely detected with or without excess σ$^{70}$ using a control template carrying base-pair substitutions that disrupt sequence-specific recognition of the transcribed-region −10-like element by σ$^{70}$ region 2 (*Deighan et al., 2011*) (*Figure 1A*). We conclude that pausing of the TEC under the control of a −10-like element within a transcribed region is sensitive to the intracellular concentration of σ$^{70}$, suggesting that σ$^{70}$ can access the TEC *in trans*, in vivo.

Next, we investigated whether or not σ$^{70}$ *trans* loading could augment the effect of a promoter-proximal pause element on the σ$^{70}$ content of downstream TECs. To do this, we used LNA probe-hybridization to detect transcripts produced from the template shown in *Figure 1B*. This λP$_{R'}$ template bears the same −10-like element as the template shown in *Figure 1A*, but in addition carries a promoter-proximal −10-like element (positioned between +1 and +6) that induces σ$^{70}$-dependent pausing at a nascent RNA length of ∼16 nt. Consistent with previous findings (*Deighan et al., 2011*), the presence of the promoter-proximal −10-like element resulted in a substantial increase (∼ninefold) in the relative abundance of the ∼35-nt pause species (compare *Figure 1A,B*). Nonetheless, when σ$^{70}$ was overproduced, the relative abundance of the ∼35-nt pause species increased further (∼1.5 fold;

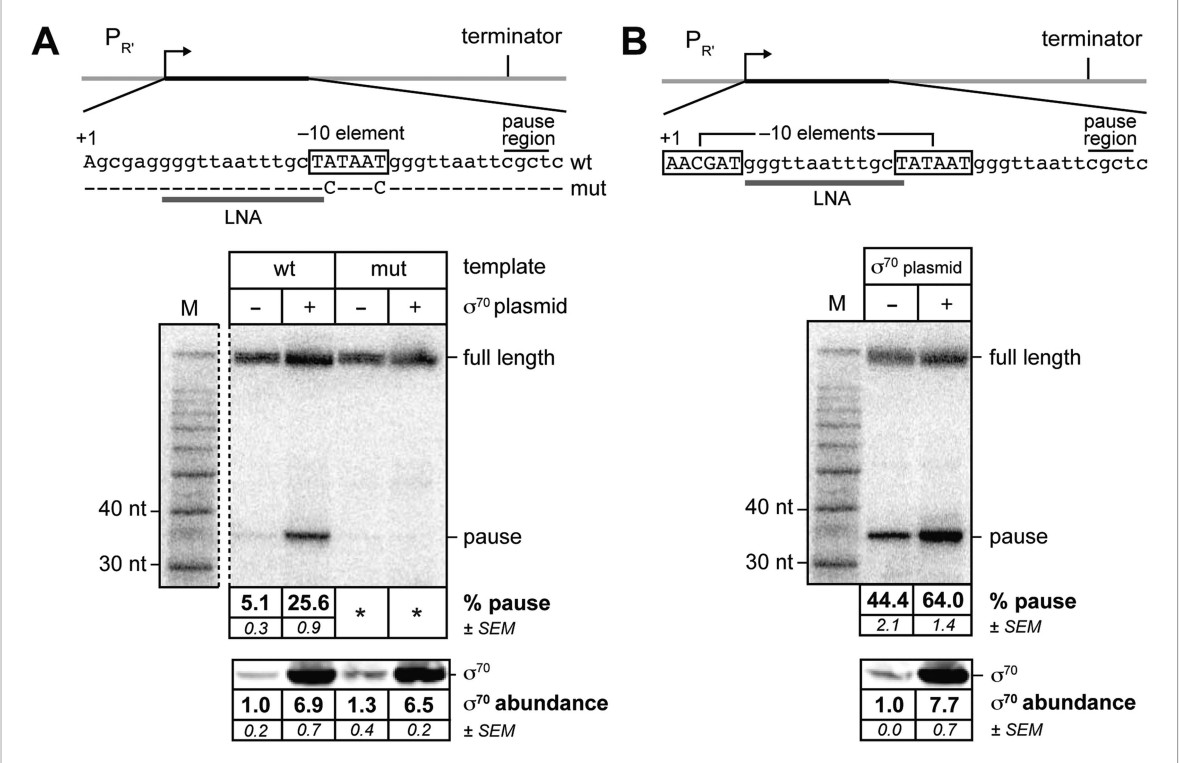

**Figure 1**. σ70 *trans* loading on a σ70-dependent transcription unit in vivo (MG1655). (**A**) *Top:* schematic of DNA template carrying λPR', transcribed-region consensus extended −10 element (wild-type or mutant) and terminator (see 'Materials and methods' for the λPR' promoter sequence). Transcribed-region sequences that are complementary to the LNA probe are underlined (grey bar) and the positions corresponding to pause sites are indicated. *middle* Analysis of RNA transcripts in vivo by LNA probe-hybridization. RNA was isolated from MG1655 cells harvested at an $OD_{600}$ of 0.8–1.0 (see 'Materials and methods'). Pausing is quantified by dividing the signal in the ~35-nt pause RNA band by the sum of this signal and the signal in the terminated (full-length) band; this ratio is expressed as a percentage (relative abundance). Mean and SEM of six independent measurements are shown. Asterisks (*) designate values that were too low (<approximately threefold above background) for accurate quantification. M, 10-nt RNA ladder. *bottom* Analysis of σ70 levels by Western blot. Amount of soluble σ70 is normalized to the amount in cells carrying the experimental template (wt) and a vector that does not direct σ70 over-production. Mean and SEM of three independent measurements are shown. (**B**) *Top*: schematic of DNA template carrying λPR', initial-transcribed-region σ70-dependent pause element, transcribed-region consensus −10 element and terminator. *middle* Analysis of RNA transcripts in vivo by locked-nucleic-acid (LNA) probe-hybridization, as in panel **A**. *bottom* Analysis of σ70 levels by Western blot.

*Figure 1B*, middle and bottom panels), indicating that the effect of the promoter-proximal −10-like element on the σ70 content of downstream TECs is not saturating.

## Detection of σ70 *trans* loading on a transcription unit expressed under the control of an alternative sigma factor in vitro and in vivo

We next sought to determine whether or not free σ70 can bind to TECs on a transcription unit controlled by an alternative σ factor. To address this possibility we generated a new template that carried a promoter recognized by RNAP holoenzyme carrying σ28, an alternative σ factor that controls the expression of genes involved in flagellar synthesis (*Chilcott and Hughes, 2000*; *Koo et al., 2009a*). This σ28 dependent promoter (P*tar*) was fused to the same transcribed region sequences that are present on the λPR' template shown in *Figure 1A* starting at position +6 (including the −10-like element; *Figure 2A*). We first performed in vitro transcription experiments to determine whether or not σ70 could access the TEC and induce pausing on this template. We formed open complexes on P*tar* using RNAP holoenzyme containing σ28 and then allowed a single round of transcription to occur in the absence or presence of excess σ70. We monitored the RNA content of each reaction at three time points after the initiation of transcription. Addition of σ70 to the transcription reactions resulted in the appearance of a cluster of RNAs ~35-nt in length (*Figure 2*, compare lanes 4–6 with lanes 1–3).

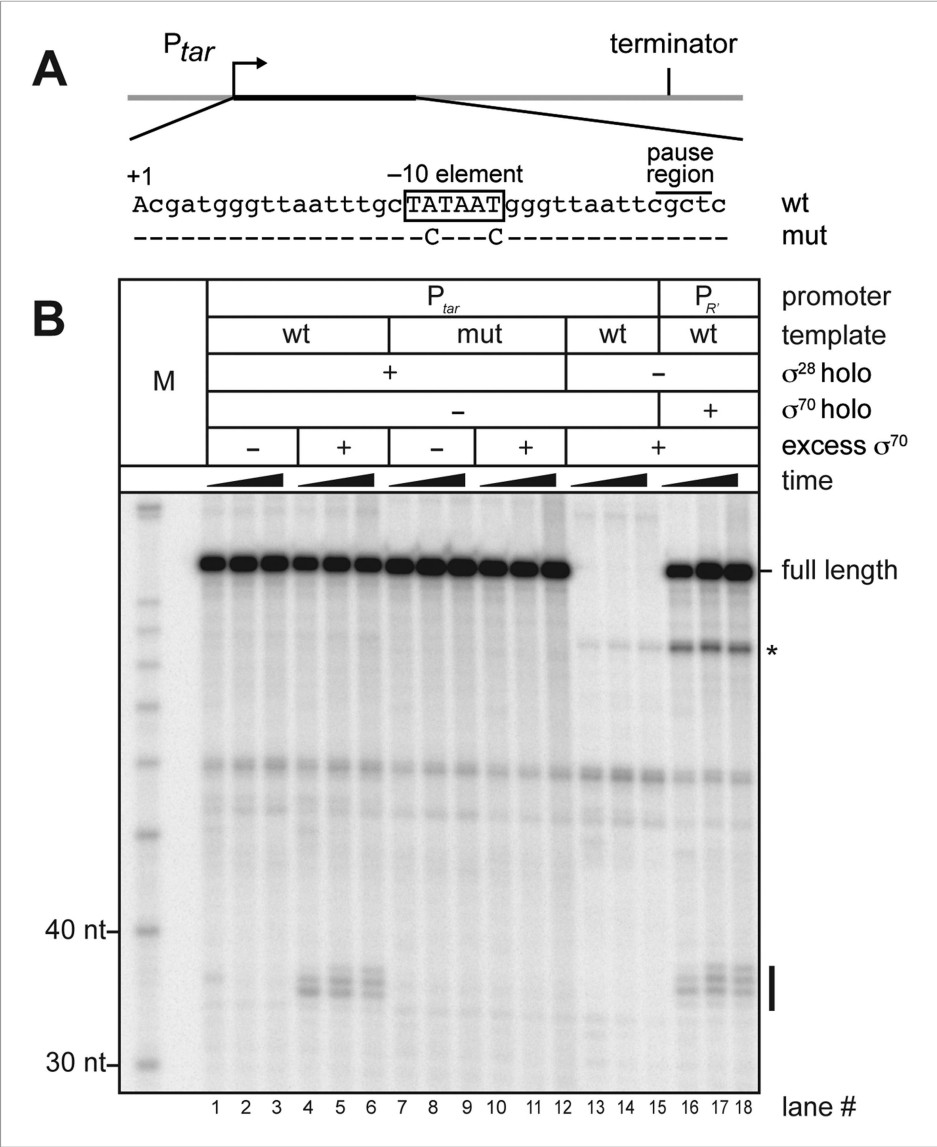

**Figure 2**. σ[70] *trans* loading on a σ[28]-dependent transcription unit in vitro. (**A**). Schematic of DNA template carrying P*tar*, transcribed-region consensus −10 element (wild-type or mutant) and terminator. Template positions corresponding to pause sites are indicated. Note that the pause sites and terminated transcripts emanating from the P*tar* promoter are located one base closer to the transcription start site (+1) than on the λP$_{R'}$ template (*Figure 1A*). (See 'Materials and methods' for the P*tar* promoter sequence.) (**B**). Analysis of RNA transcripts in vitro. Single-round in vitro transcription reactions were performed with reconstituted RNA polymerase (RNAP) holoenzyme containing σ[28] (lanes 1–12), RNAP core enzyme (lanes 13–15) or reconstituted RNAP holoenzyme containing σ[70] (lanes 16–18) and three different templates: P*tar* with a wild-type (wt) transcribed-region −10 element (lanes 1–6 & 13–15) or a mutated (mut) transcribed-region −10 element (lanes 7–12) and λP$_{R'}$ with a wild-type transcribed-region −10 element (lanes 16–18). The reactions were performed as a time course with samples taken at 1, 6 and 18 min after transcription was initiated; these reactions were performed in the absence of transcript cleavage factors GreA and GreB, resulting in a characteristic pattern of long-lived pause species (*Deighan et al., 2011*). Where indicated, excess σ[70] (1 µM) was added with the 'start mix' after open complex formation. RNAs associated with paused transcription elongation complexes (TECs) (pause) and terminated transcripts (full length) are labeled. The asterisk (*) indicates a shorter terminated transcript that is the result of transcription initiating under the control of the transcribed-region −10 element when the σ[70]-containing holoenzyme is present in the reaction.

These RNAs were not observed when reactions were performed using a control template carrying disruptive base-pair substitutions within the transcribed-region −10-like element (*Figure 2B*, lanes 7–12). A set of reactions performed in the presence of σ70 but in the absence of σ28 confirmed that appearance of the cluster of ~35-nt RNAs is strictly dependent on transcription that initiates from P*tar* under the control of σ28 (*Figure 2B*, lanes 13–15). In addition, the distribution of RNA species within this cluster closely resembles that within a similar cluster produced when reactions were performed using the λP_{R'} template (*Figure 1A*) and RNAP holoenzyme containing σ70 (*Figure 2B*, lanes 16–18). We conclude that the ~35-nt RNAs are pause RNAs that arise due to the ability of σ70 to bind TECs generated via transcription initiating at P*tar* under the control of σ28. These findings therefore indicate that free σ70 can bind to TECs on a σ28-controlled transcription unit in vitro.

We then sought to determine whether or not σ70 can bind to TECs on a σ28-controlled transcription unit in vivo. For this experiment we introduced into cells three compatible plasmids. The first plasmid carried either the wild-type P*tar* template or a mutant P*tar* template with base-pair substitutions that disrupt sequence-specific recognition of the transcribed-region −10-like element by σ70. The second plasmid did or did not direct the production of excess σ70 and the third plasmid did or did not direct the production of excess σ28. We isolated total RNA and soluble protein from cells and used LNA probe-hybridization to detect transcripts emanating from the P*tar* promoter (*Figure 3A*, top) and Western blotting to assess the concentrations of σ70 (*Figure 3A*, middle) and σ28 (*Figure 3A*, bottom).

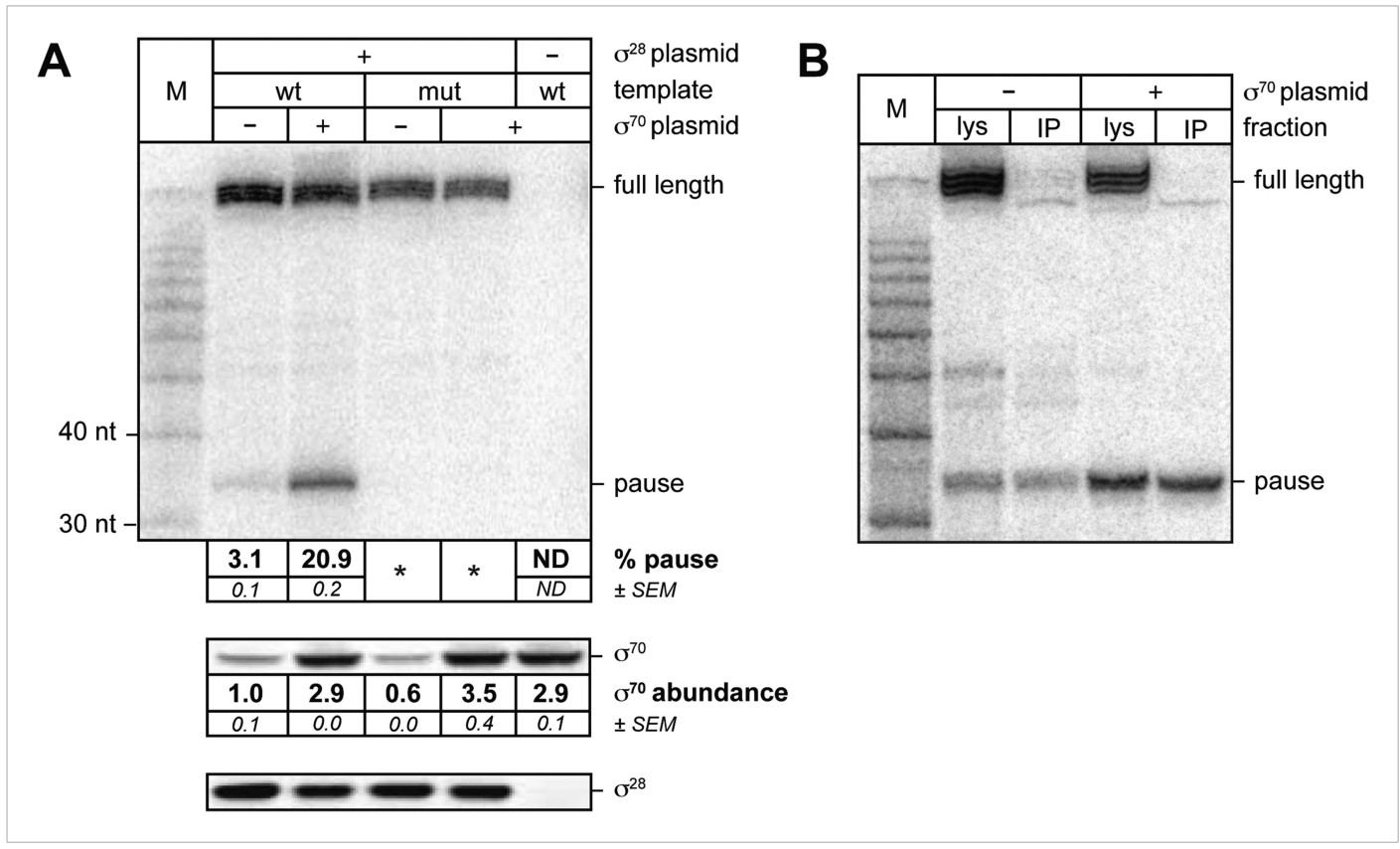

**Figure 3**. σ70 *trans* loading on a σ28-dependent transcription unit in vivo. (**A**). *top* Detection of RNA transcripts in vivo from the templates shown in *Figure 2A* by LNA probe-hybridization. Transcribed-region sequences that are complementary to the LNA probe are as in *Figure 1A*. RNA was isolated from MG1655 cells harvested at an OD_{600} of 0.8–1.0. Pausing is quantified by dividing the signal in the ~35-nt pause RNA band by the sum of this signal and the signal in the terminated (full-length) band. Mean and SEM of three independent measurements are shown. Asterisks (*) designate values that were too low for accurate quantification. M, 10-nt RNA ladder. *middle* Analysis of σ70 levels by Western blot. Amount of soluble σ70 is normalized to the amount in cells carrying the experimental template (wt) and a vector that does not direct σ70 over-production. Mean and SEM of three independent measurements are shown. *bottom* Analysis of σ28 levels by Western blot. (**B**). Analysis of RNAP-associated transcripts produced from the wild-type P*tar* template. RNA was isolated from the lysate fraction (lys) or the immunoprecipitated fraction (IP) of SG110 cells (OD_{600} ~0.5) and analyzed by LNA probe-hybridization. The cells contained a vector directing the synthesis of σ28, as well as a vector that did or did not direct σ70 overproduction.

We found that transcripts emanating from P*tar* were detected only in cells carrying the plasmid that directed the synthesis of excess σ28 (*Figure 3A*, compare lanes 2–5 with lane 6). Furthermore, in the presence of excess σ28 but in the absence of excess σ70, we detected a small amount of an RNA species that migrated between the 30-nt and 40-nt RNA markers (*Figure 3A*, lane 2). This RNA species was similar in size to the ~35-nt pause RNA detected by LNA probe-hybridization with the λP_R′ template in vivo (*Figure 1A*) and to the cluster of ~35-nt pause RNAs produced from the P*tar* template in vitro in the presence of excess σ70 (*Figure 2B*, lanes 4–6). We found that the relative abundance of this ~35-nt RNA was increased ~sevenfold when σ70 was overproduced by a factor of ~3 (*Figure 3A*, compare lanes 2 and 3). In addition, the ~35-nt RNA was not detected in cells containing the mutant P*tar* template carrying base-pair substitutions in the transcribed-region −10-like element (*Figure 3A*, lanes 4 and 5).

Next, we sought to determine whether or not the ~35-nt RNA species produced under the control of the P*tar* promoter was RNAP-associated, as would be expected for a pause RNA. To carry out this experiment, we used a strain carrying a chromosomal *rpoC-3xFLAG* gene, encoding the RNAP β′ subunit with a C-terminal 3xFLAG tag, which enables us to isolate RNAP-associated transcripts by immunoprecipitating RNAP with an antibody against FLAG. We introduced into this strain the plasmid carrying the wild-type P*tar* template, the plasmid directing the production of excess σ70 or the corresponding empty vector, and the plasmid directing the production of excess σ28. We isolated RNA from cell lysates (*Figure 3B*, lys) or from 3xFLAG-tagged TECs immunoprecipitated with an antibody against FLAG (*Figure 3B*, IP) and used LNA probe-hybridization to detect transcripts emanating from the P*tar* promoter. The results indicate that a major fraction of the ~35-nt RNA species, but not the full-length terminated transcript, was immunoprecipitated with an antibody against FLAG whether the cells lacked or contained plasmid encoded overproduced σ70 (*Figure 3B*). Thus, we conclude that a major fraction of the ~35-nt RNA species, but not the full-length terminated transcript, is RNAP-associated.

Taken together, the results of *Figure 3* establish that the appearance of the ~35-nt RNA depends both on the presence of σ28 and on an intact −10-like element, that the relative abundance of the ~35-nt RNA is increased upon overproduction of σ70, and that the ~35-nt RNA is RNAP-associated. We therefore conclude that the ~35-nt RNA produced from the P*tar* template in vivo represents a pause RNA that arises due to the ability of σ70 to bind TECs generated under the control of σ28. Furthermore, our ability to detect σ70-dependent pause species produced under the control of a promoter that is recognized by an alternative σ factor enables us unambiguously to identify pausing that is mediated by *trans*-loaded σ70. Thus, our findings establish that σ70 can access the TEC *in trans*, in vivo.

## Effect of growth phase on the extent of σ70-dependent pausing due to *trans* loaded σ70

Although experiments using the P*tar* template revealed that σ70 *trans* loading is detectable even in the absence of σ70 overproduction, we found that during the exponential phase of growth the extent of pausing due to chromosomally encoded *trans*-loaded σ70 appeared to be low (*Figures 3A* and *4A*; the relative abundance of the ~35-nt RNA was <5%). However, when we harvested RNA from stationary phase cells containing the P*tar* template, we found that the relative abundance of the ~35-nt RNA was ~50% (*Figure 4A*, lane 3), which was reduced to ~10% when the transcribed-region −10-like element was mutated (*Figure 4A*, lane 5). Furthermore, like those detected during exponential phase, the ~35-nt RNAs detected from the P*tar* template during stationary phase were RNAP-associated (*Figure 4—figure supplement 1A*). Thus, the ~35-nt RNAs detected during both exponential phase and stationary phase exhibit hallmarks of a σ70-dependent pause species (stable association with RNAP and sensitivity to mutations in the transcribed region −10-like element). We conclude that the relative abundance of pause RNAs that arise due to σ70 *trans* loading varies with growth-phase.

To investigate the basis for the growth phase-dependent change in the abundance of the pause RNAs that arise due to σ70 *trans* loading, we first performed Western blot analysis to compare the amounts of σ70 in exponential and stationary phase cells. The results indicated that the cell extracts prepared from exponential and stationary phase cultures contained comparable amounts of σ70 (*Figure 4—figure supplement 1B*). We conclude that the growth phase-dependent increase in the abundance of the ~35-nt pause RNAs is not a consequence of an increase in the total cellular concentration of σ70. (We note that these data do not exclude the possibility that growth

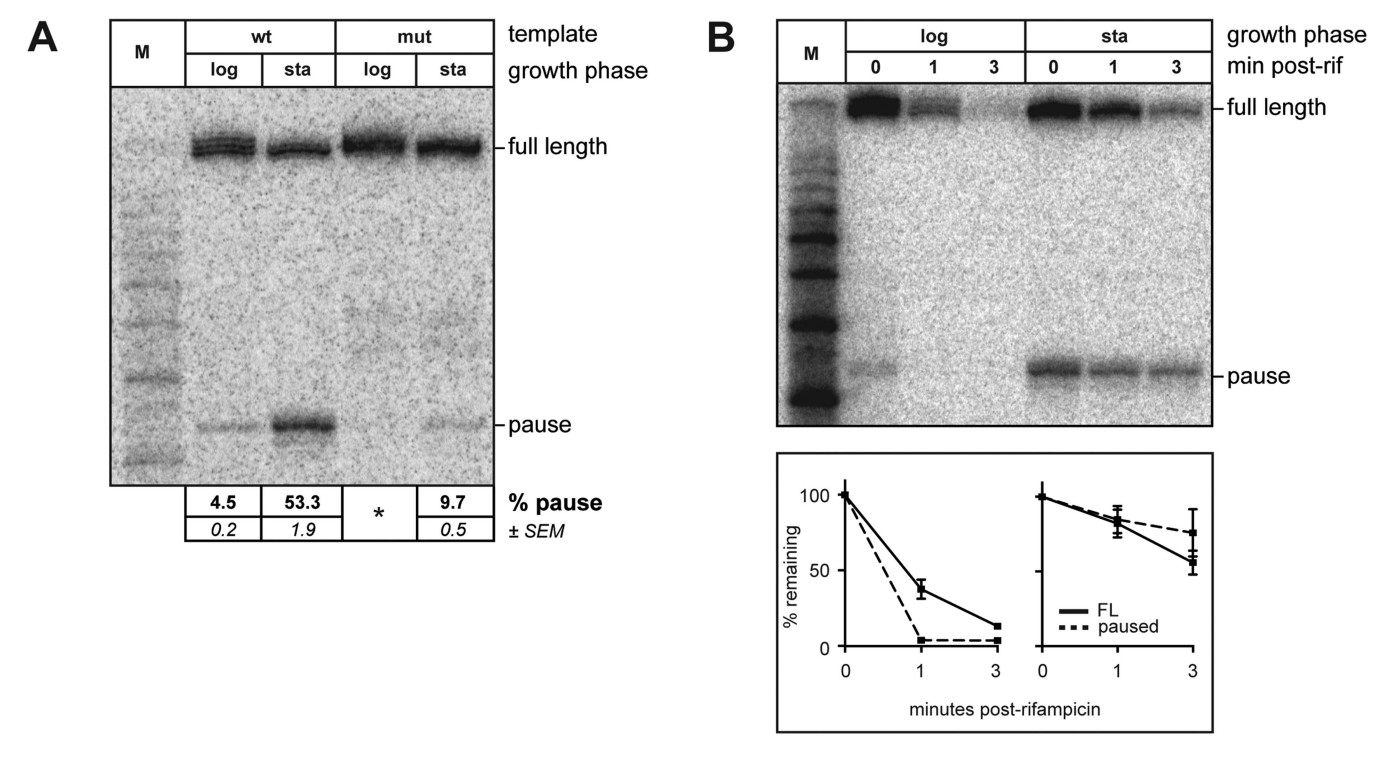

**Figure 4**. Growth phase dependent σ$^{70}$ *trans* loading on a σ$^{28}$-dependent transcription unit in vivo. (**A**). Detection of RNA transcripts in vivo from the templates shown in *Figure 2A* by LNA probe-hybridization. Transcribed-region sequences that are complementary to the LNA probe are as in *Figure 1A*. RNA was isolated from SG110 cells harvested at an OD$_{600}$ of ~0.5 (log) or ~2.5 (sta). Pausing is quantified by dividing the signal in the ~35-nt pause RNA band by the sum of this signal and the signal in the terminated (full-length) band. Mean and SEM of six independent measurements are shown. Asterisks (*) designate values that were too low for accurate quantification. M, 10-nt RNA ladder. (**B**). *top* Detection of RNA transcripts derived from the wt template in vivo after treatment with rifampicin. *bottom* Percent of transcript remaining relative to T = 0 at indicated time points after addition of rifampicin. Mean and SEM of ten (log, 1 m), eight (sta, 1 m), or six (log and sta, 3 m) independent measurements are shown.

The following figure supplement is available for figure 4:

**Figure supplement 1**. (**A**). Analysis of RNAP-associated transcripts produced from the wild-type P*tar* template.

phase-dependent changes in the amount of free σ$^{70}$ available to bind the TEC contribute to changes in the abundance of the pause RNAs that arise due to σ$^{70}$ *trans* loading.)

We next used the RNAP inhibitor rifampicin to analyze the half-life of pause RNAs that arise due to σ$^{70}$ *trans* loading during exponential phase or stationary phase. To do this, we isolated RNA from cells either just before or 1 and 3 min after rifampicin treatment and used LNA-probe hybridization to measure the decay of the ~35-nt RNAs and full-length transcripts. We found that the half-life of the ~35-nt pause RNA was greater in stationary phase than in exponential phase (*Figure 4B*). In addition, the full-length terminated transcript was at least as stable in stationary phase as in exponential phase (*Figure 4B*), excluding the possibility that the increase in the relative abundance of the pause RNA might simply reflect a destabilization of the full-length transcript in stationary phase. Thus, our findings indicate that the extent of pausing on the P*tar* template due to *trans*-loaded σ$^{70}$ varies with growth phase, at least in part, due to an increase in the half-life of the pause.

## Discussion

Here we show that the primary σ factor of *E. coli* can act as a classical elongation factor and engage the TEC *in trans*, in vivo, inducing transcription pausing by binding transcribed-region promoter-like elements (*Figures 1, 3 and 4*). Furthermore, we find that the extent of pausing due to *trans*-loaded σ$^{70}$

varies with growth phase, becoming most prominent during the stationary phase of growth (*Figure 4*). We demonstrate that σ⁷⁰ *trans* loading can occur in vivo regardless of whether the TEC was generated through initiation at a σ⁷⁰-dependent promoter (*Figure 1*) or a promoter that is recognized by an alternative σ factor (*Figures 3, 4*). Our findings indicate that at least two distinct σ factors can influence the functional properties of a transcription complex during the transcription cycle in vivo: one during initiation and one (or more) during elongation.

## Dual pathways for σ⁷⁰ to associate with the TEC in vivo

The results presented here, coupled with prior work (*Shimamoto et al., 1986*; *Ring et al., 1996*; *Bar-Nahum and Nudler, 2001*; *Mukhopadhyay et al., 2001*; *Brodolin et al., 2004*; *Nickels et al., 2004*; *Wade and Struhl, 2004*; *Kapanidis et al., 2005*; *Raffaelle et al., 2005*; *Reppas et al., 2006*; *Mooney et al., 2009*; *Deighan et al., 2011*) define two pathways whereby σ⁷⁰ can access the TEC in vivo, a pathway that operates in *cis* and a pathway that operates in *trans* (*Figure 5*). The *cis*-acting pathway depends on retention in the TEC of the σ⁷⁰ that was used during initiation, with the extent of σ⁷⁰ retention being modulated by the sequence of the initial transcribed region (*Figure 5A*) (*Deighan et al., 2011*). Thus, the *cis*-acting (retention) pathway is necessarily restricted to transcription units controlled by σ⁷⁰-dependent promoters. In contrast, the *trans*-acting pathway identified in this study, which can be functionally defined by its sensitivity to the intracellular concentration of σ⁷⁰, is potentially operative on all transcription units (*Figure 5B*). Moreover, the two pathways can function in concert. Thus, we found that σ⁷⁰ *trans* loading can increase the σ⁷⁰ content of TECs generated under the control of a σ⁷⁰-dependent promoter even in the presence of an initial-transcribed-region σ⁷⁰-dependent pause element that augments σ⁷⁰ retention.

The use of a transcription unit expressed under the control of an alternative σ factor enabled us to analyze the *trans*-acting pathway independent of the *cis*-acting pathway. We found that the effects

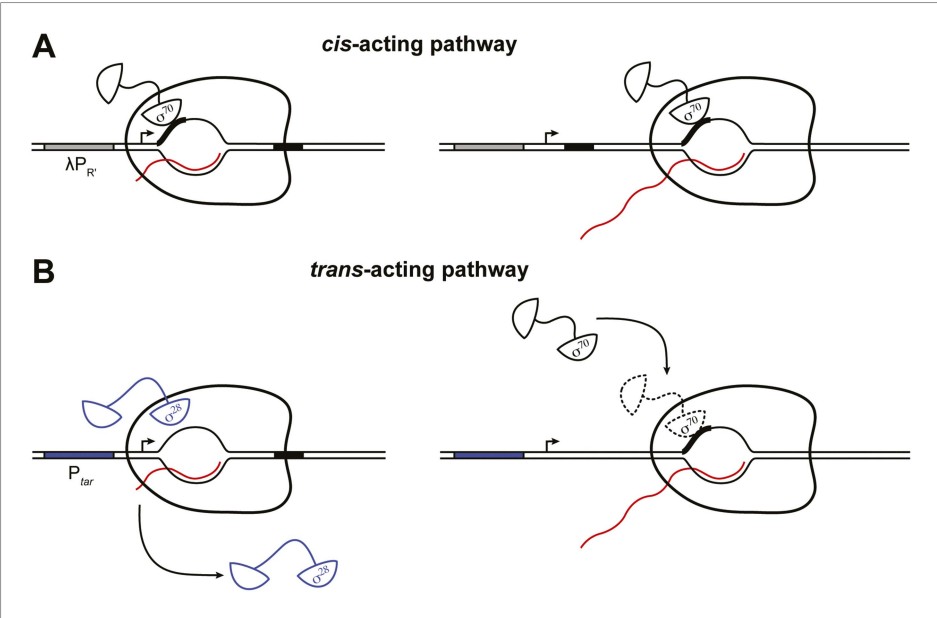

**Figure 5**. Dual pathways for σ⁷⁰ to associate with the TEC in vivo. (**A**). *Cis*-acting pathway (*Deighan et al., 2011*). The retention in the TEC of the σ⁷⁰ that was used during initiation enables pausing at transcribed-region −10-like elements on transcription units that are expressed under the control of σ⁷⁰-dependent promoters. Presence of an initial-transcribed-region σ⁷⁰-dependent −10-like element increases the σ⁷⁰ content of downstream TECs and increases the efficiency of pausing at a second σ⁷⁰-dependent pause element further downstream. Promoter, grey rectangle; σ⁷⁰-dependent pause elements, black rectangles; RNA, wavy red line. (**B**). *Trans*-acting pathway. Binding of σ⁷⁰ to TECs that have lost the σ factor used during initiation (here, σ²⁸) increases the efficiency of pausing at a transcribed-region σ⁷⁰-dependent pause element. Promoter, blue rectangle; σ⁷⁰-dependent pause element, black rectangle; RNA, wavy red line.

of *trans*-loaded $\sigma^{70}$ on pausing varied with growth phase. In particular, pausing mediated by chromosomally encoded *trans*-loaded $\sigma^{70}$ was detectable during the exponential phase of growth and this pausing became more prominent during stationary phase. Our experiments revealed that this increase in the relative abundance of the pause species during stationary phase was explained at least in part by an increase in pause half-life in stationary phase cells as compared to exponentially growing cells. We speculate that this increase in pause half-life might be due to a drop in the intracellular NTP concentrations as nutrients are depleted and the cells enter stationary phase (*Buckstein et al., 2008*).

It is intriguing to consider our findings in light of a prior report of growth-phase dependent changes in the ability of purified RNAP holoenzyme to retain $\sigma^{70}$ during transcription elongation as assayed in vitro (*Bar-Nahum and Nudler, 2001*). In particular, the authors of this study found that RNAP holoenzyme purified from stationary phase cells produced a substantially higher fraction of $\sigma^{70}$-containing TECs than did RNAP holoenzyme purified from exponentially growing cells, possibly suggesting that the stationary phase RNAP core enzyme binds $\sigma^{70}$ more tightly.

We note that the effect of growth phase on the relative abundance of pause RNAs may not be limited to $\sigma^{70}$–dependent pausing. In fact, our experiments revealed a potential pause species that was detectable above background in stationary phase cells even when the transcribed-region −10-like element was mutated (*Figure 4A*, lane 5). We suggest that this RNA arises due to the presence of an overlapping consensus pause element that is recognized by the core RNAP ($G_{-10}Y_{-1}G_{+1}$; see *Figure 2A*) (*Herbert et al., 2006*; *Larson et al., 2014*; *Vvedenskaya et al., 2014*) and is not disrupted by the mutations in the −10-like element.

### σ cross regulation

Two or more different σ factors often control the expression of a given gene by directing initiation from distinct upstream promoters (*Wade et al., 2006*; *Gama-Castro et al., 2008*; *Cho et al., 2014*). Our findings illustrate another mechanism whereby the combined input of multiple σ factors can modulate gene expression. Specifically, we show that distinct σ factors can direct initiation and modulate elongation on the same transcription unit. Such 'σ cross regulation' might enable the cell to integrate signals transmitted via $\sigma^{70}$ and an alternative σ factor to modulate gene expression within a single transcription unit under the control of a non $\sigma^{70}$–dependent promoter. In principle there are several ways that $\sigma^{70}$ *trans* loading might modulate gene expression. First, as shown here, $\sigma^{70}$ *trans* loading can cause the TEC to pause, which is expected to influence transcription output directly in a manner that depends on pause half-life (and, as suggested by our results shown in *Figure 4*, may become particularly relevant in stationary phase). $\sigma^{70}$–dependent pausing might also influence gene expression indirectly, by facilitating engagement of regulatory factors, influencing formation of RNA secondary structures, or influencing translation (*Roberts et al., 1998*; *Wickiser et al., 2005*; *Landick, 2006*; *Pan and Sosnick, 2006*; *Lemay et al., 2011*; *Perdrizet et al., 2012*; *Larson et al., 2014*; *Nechooshtan et al., 2014*; *Belogurov and Artsimovitch, 2015*). Second, $\sigma^{70}$ *trans* loading might impede the accessibility of the TEC to other elongation factors such as NusG or RfaH, which share the same primary binding site on RNAP (*Sevostyanova et al., 2008*; *Mooney et al., 2009*).

Future work will be required to investigate the extent to which $\sigma^{70}$ *trans* loading contributes to gene expression through these or other mechanisms. In this regard, the application of sequencing-based methodologies such as native elongating transcript sequencing (NET-seq) (*Churchman and Weissman, 2011*; *Larson et al., 2014*; *Vvedenskaya et al., 2014*) and chromatin immunoprecipitation sequencing (ChIP-seq) (*Myers et al., 2015*) should enable the identification of transcription units that manifest growth phase-dependent pausing attributable to *trans* loaded $\sigma^{70}$. Nevertheless, our findings add to a growing body of evidence that the functions of σ are not limited to the initiation phase of transcription. Furthermore, they establish a mechanism whereby the primary σ factor can extend its reach by exerting direct effects on the composition of the entire transcriptome, not just that portion that is produced under the control of $\sigma^{70}$-dependent promoters.

## Materials and methods

### Strains

All experiments were performed in *E. coli* strain MG1655 or SG110 (*Vvedenskaya et al., 2014*) in which the chromosomal *rpoC* gene is fused to a 3xFLAG epitope tag-encoding sequence.

## Plasmids

Plasmids used in this study are listed in *Table 1*. Promoter sequences are as follows. λP$_{R'}$ : TTGACT tattgaataaaattgggTAAATTtgactcA and P*tar*: TAAAGTTTcccccctccttGCCGATAAcgagatcA, where the −10 and −35 elements and the +1 nucleotide are capitalized.

## Cell growth

Single colonies of *E. coli* strains bearing the appropriate plasmids were used to inoculate individual 5 ml aliquots of LB broth (Miller) (10 g tryptone, 5 g yeast extract, 10 g NaCl per liter) (EMD-Millipore, Billerica, MA) containing antibiotics (spectinomyin [50 µg/ml] and streptomycin [25 µg/ml] were used together to maintain vectors bearing the *aadA1 [Sm$^R$]* allele; carbenicillin [100 µg/ml]; chloramphenicol [25 µg/ml]) in 18 × 150 mm glass culture tubes which were incubated, rolling, overnight at 37°C. Aliquots of these cultures were diluted 1:100 into 25 ml of LB containing antibiotics and IPTG (1 mM) in 125 ml DeLong flasks with Morton-style closures (Bellco Glass, Vineland, NJ), shaken at 37°C on an orbital platform shaker at 220–250 RPM. For the experiments shown in *Figure 4*, cultures were grown as described above, except that cells were initially back-diluted into a volume of 75 ml of media containing antibiotics and IPTG, mixed, and then 25 ml aliquots were transferred into each of two 125 ml flasks. One aliquot was used for each harvest time-point.

## RNA isolation

### Standard method (*Figure 1* and *Figure 3*)

When cultures reached an OD$_{600}$ between 0.8 and 1.0, 5 ml aliquots were harvested into 50 ml Oakridge tubes containing 15 ml of RNAlater solution (Life Technologies, Grand Island, NY) and mixed several times by inversion. Cell suspensions in RNAlater were incubated overnight at 4°C. Cells suspended in RNAlater were centrifuged at 17,000 × g for 20 min at 4°C; the supernatant was decanted

**Table 1.** Plasmids

| Plasmid | Description | Source |
|---|---|---|
| pLHN12-His | pT7-His$_6$-*rpoD* | (*Panaghie et al., 2000*) |
| pET15b-His-fliA | pT7-His$_6$-*fliA* | This work |
| pFW11tet-P$_{R'}$_+19 | λP$_{R'}$ promoter and native σ$^{70}$-dependent pause element with a second σ$^{70}$-dependent pause element located 19 bp downstream of the +1 transcription start site. The t$_{R'}$ intrinsic terminator is positioned to terminate transcription ~116 bp downstream of +1. | (*Deighan et al., 2011*) |
| pFW11tet-mutP$_{R'}$_+19 | Same as pFW11tet-P$_{R'}$_+19 but with A+2 G/T+6 G mutations in the native σ$^{70}$-dependent pause element. | (*Deighan et al., 2011*) |
| pFW11tet-Ptar_+19 (pNUN175) | Same as pFW11tet-P$_{R'}$_+19 except that the promoter (up to and including +1) has been replaced with the σ$^{28}$-dependent P*tar* promoter. | This work |
| pFW11tet-Ptar_mut+19 (pNUN176) | Same as pFW11tet-Ptar_+19 but with mutations in the pause element. | This work |
| pBR-*fliA* | pSG585-*fliA* | This work |
| pSG585 | colE1 origin plasmid with lacUV5 upstream of multiple cloning site | This work |
| pNUN191 | pCDFlacMUT3-rpoD | This work |
| pCDFlacMUT3 | pCDFlac with attenuated −35 element (AATACA) | This work |
| pCDFlac | derivative of pCDF-1b into which the lacUV5 promoter has been inserted | (*Montero-Diez et al., 2013*) |

and residual liquid carefully removed by pipetting. 1 ml of Tri-reagent (Molecular Research Center, Cincinnati, OH) was added to each tube and pellets were dispersed by vortexing. Cell suspensions in Tri-reagent were transferred to 1.7 ml low binding tubes (BioExcell; Worldwide Medical Products, Bristol, PA), incubated at 70°C for 10 min, centrifuged at 21,000 × g at 4°C for 10 min, and the supernatants were recovered into fresh tubes. 200 μl of chloroform was added to each tube and mixed by vigorous shaking for 15 s. Phases were separated by centrifugation at 21,000 × g at 4°C for 15 min. 500 μl of the upper, aqueous phase was recovered and transferred to a fresh tube to which 167 μl of 100% ethanol was added. Subsequent removal of RNA >200 nt and recovery of RNA <200 nt was performed using the mirVana microRNA Isolation kit (Life Technologies) according to the manufacturer's protocol. After elution from mirVana columns, eluents were concentrated by ethanol precipitation and resuspended directly into formamide loading dye (95% deionized formamide, 18 mM EDTA, and 0.025% SDS, xylene cyanol, bromophenol blue, amaranth).

## Rapid harvest method (*Figure 4*) for RNA stability measurements

At T = 0 (log phase: $OD_{600}$ ~0.5 after 3hr growth; stationary phase: $OD_{600}$ ~2.5 after 21hr growth), a 2 ml aliquot was withdrawn from each culture and transferred to a 2 ml microcentrifuge tube and cells were immediately pelleted by centrifugation at 10,000 × g for 30 s at 37°C. Supernatants were decanted and pellets placed immediately onto dry ice. To facilitate rapid handling, the dry ice and microcentrifuge were placed adjacent to the platform shaker in the 37°C environmental room and the microcentrifuge was allowed to equilibrate to ambient temperature prior to use. After the T = 0 pellets were frozen, rifampicin (50 mg/ml in DMSO) was added to the remaining culture volume to a final concentration of 1 mg/ml. Rapid harvest was performed as described for the T = 0 fraction. The post-rifampicin time-points refer to when the cell pellets were placed on dry ice. Pellets were stored at −80°C until needed. Frozen cell pellets were resuspended directly into 200 μl of Tri-reagent, heated at 60°C for 10 min, cleared of debris by centrifugation at 21,000 × g at 4°C for 10 min, and the supernatants were recovered into fresh tubes. 40 μl of chloroform was added to each tube and mixed by vigorous shaking for 15 s. Phases were separated by centrifugation at 21,000 × g at 4°C for 15 min and 100 μl of the upper, aqueous phase was recovered. 33 μl of 100% ethanol was mixed with the recovered aqueous phase, the mixture was applied to a mirVana spin cartridge, and flowthrough collected after centrifugation at 10,000 × g at room temperature for 1 min. RNA was precipitated from the flowthrough by addition of 1 μl of 10 mg/ml glycogen and 240 μl of 100% ethanol followed by incubation at −20°C for 12–18 hr. Pellets were resuspended directly into formamide loading dye (see above).

## Method for isolation of RNAP-associated RNA (*Figure 3B* and *Figure 4—figure supplement 1B*)

After cell cultures reached an $OD_{600}$ of ~0.5 (log phase) or ~2.5 (stationary phase), a 10 ml volume of cell culture was centrifuged at 8000 × g for 5 min at ambient temperature. Supernatants were decanted and pellets frozen on dry ice. Cell lysis and RNA isolation were performed as described by *Vvedenskaya et al. (2014)*.

## LNA probe labeling

50 pmol of LNA probe (5′ agCaaAttAacCc 3′), where LNA bases are capitalized; Exiqon, Woburn, MA) was incubated in a 25 μl volume with 5 μl γ-$^{32}$P-ATP (EasyTide; Perkin Elmer, Waltham, MA), 2.5 μl 10X T4 PNK buffer, 13.5 μl nuclease free water (Life Technologies), and 2 μl T4 PNK (NEB, Ipswich, MA) at 37°C for 1 hr followed by 95°C for 10 min. Labeled probe was separated from unincorporated radiolabeled nucleotide using a size-exclusion spin column (SigmaSpin; Sigma–Aldrich, St. Louis, MO).

## Detection of pause RNAs and full-length RNAs in vivo by LNA hybridization

RNAs generated in vivo were detected by hybridization as described in (*Pall and Hamilton, 2008*; *Goldman et al., 2009*; *Deighan et al., 2011*) using a 5′ radiolabeled LNA probe. RNA was loaded onto 0.4 mm thick 20% denaturing polyacrylamide slab gels cast and equilibrated in 50 mM MOPS (pH 7 with NaOH), transferred to neutral nylon membrane (Whatman Nytran N; GE Healthcare Life Sciences, Piscataway, NJ) using a semi-dry electroblotting apparatus (Biorad, Hercules, CA) operating at 20V for 25 min using chilled 20 mM MOPS (pH 7 with NaOH) as conductive medium. RNA was crosslinked to the membrane using 157 mM *N*-(3-dimethylaminopropyl)-*N*′-ethylcarbodiimide

hydrochloride (EDC) (Sigma–Aldrich) in 0.97% 1-methylimidazole (pH 8) (Alfa Aesar, Ward Hill, MA) (as described in *Pall and Hamilton, 2008*) for 80 min at 55°C. Crosslinking solution was rinsed from the membrane by immersion in 20 mM MOPS (pH 7 with NaOH) at 25°C, the membrane was placed onto nylon hybridization mesh, the membrane-mesh stack was placed into a 70 × 150 mm hybridization bottle at 50°C and 50 ml of pre-hybridization solution (5× SSC, 5% SDS, 2× Denhardt's solution, 40 µg/ml sheared salmon sperm DNA solution [Life Technologies], 20 mM $Na_2HPO_4$ [pH 7.2] in diethylpyro-carbonate (DEPC) treated water) at 50°C was added. The hybridization bottle was rotated in a hybrization oven at 50°C for 30 min, the solution was decanted and replaced by a 50 ml portion of pre-warmed hybridization solution that had been thoroughly mixed with the entire volume of the radiolabeled LNA probe prepared above. The bottle was then returned to the 50°C oven for 16 hr. The membrane was washed twice for 10 min in non-stringent wash buffer (3× SSC, 5% SDS, 10× Denhardt's solution, 20 mM $Na_2HPO_4$ [pH 7.2] in DEPC treated water), twice for 30 min in non-stringent wash buffer, and once for 5 min in stringent wash buffer (1× SSC, 1% SDS, in DEPC treated water) before it was blotted dry, wrapped in plastic film, and radiolabeled bands were visualized by storage phosphor screen (GE Healthcare) and phosphorimagery (Storm 830 imager or Typhoon 9400 variable mode imager, GE Healthcare). All wash buffers were equilibrated to 55°C prior to use. Hybridization oven was operated at 50°C throughout.

## Protein isolation for immunoblotting

With the exception of *Figure 4—figure supplement 1B*, protein isolation for immunoblotting was performed as follows. 1 ml of cell suspensions was pelleted by centrifugation at 10,000 × g for 2 min at ambient temperature, supernatants were carefully removed by vacuum aspiration and pellets were immediately frozen on dry ice before being stored at −80°C. To extract soluble protein, cell pellets were thawed on ice for ∼30 s and resuspended by pipetting in lysis solution normalized to 50 µl per 1 ml of $OD_{600} = 0.6$. Lysis solution consisted of 1 ml B-PER reagent (Thermo Scientific Pierce, Rockland, IL), 1/4 protease inhibitor tablet (Comlete-mini [EDTA-free]; Roche, Indianapolis, IN), 2 µl 0.5M EDTA (pH 8), 2 µl lysozyme (50 mg/ml), 120 µl TurboDNase (Life Technologies), and 200 µl 10× TurboDNase buffer. Lysis mixture was incubated 10 min on ice. Lysates were centrifuged at 21,000 × g for 10 min at 4°C to pellet insoluble material. 40 µl of clarified supernatant was then mixed with an equal volume of 2× loading buffer prepared by mixing 500 µl 4× NuPAGE LDS sample buffer (Life Technologies), 50 µl β-mercaptoethanol and 450 µl water. Samples were heated at 70°C for 2 min and centrifuged at 21,000 × g for 2 min at ambient temperature prior to electrophoresis.

For the experiment of *Figure 4—figure supplement 1B*, total cellular protein was isolated as follows. Cell pellets, obtained and stored as described above, were resuspended directly into 50 µl per 1 ml of $OD_{600} = 0.6$ of 1× Laemmli SDS sample buffer (pH 7.4) and heated 90°C for 5 min. Debris was pelleted by centrifugation at 21,000 × g for 5 min and the supernatants were transferred to fresh tubes and analyzed by gel electrophoresis.

## Immunoblotting

With the exception of *Figure 4—figure supplement 1B*, immunoblots were performed as follows. 10 µl of each soluble protein sample was loaded onto a 4–12% gradient NuPAGE Novex Bis-Tris precast gel (Life Technologies) and run in 1X NuPAGE MOPS SDS running buffer until the dye front exited the gel. The gel cassette was then opened and the gel was equilibrated into transfer buffer (192 mM glycine, 25 mM Tris, 10% methanol) for 5–10 min. PVDF membrane (Immobilon-FL; EMD-Millipore) was wetted in 100% methanol and equilibrated into transfer buffer prior to transfer-stack assembly. Semi-dry electro transfer was performed using a Trans-Blot SD apparatus (Bio-Rad) operating at 10V for 1 hr. After transfer, membranes were placed into blocking solution (5% non-fat dry milk in 1× PBS) and gently agitated at ambient temperature for 30 min. Blocking solution was decanted and replaced with 10 ml of a 1:5000 dilution of affinity purified mouse monoclonal antibody recognizing $\sigma^{70}$ (clone 2G10; Neoclone, Madison, WI) or $\sigma^{28}$ (clone 1RF18; Neoclone) in blocking solution and gently agitated for 1 hr as above. The primary antibody solution was decanted and the membrane washed quickly in 10 sequential portions of blocking solution containing 0.1% TWEEN-20. Goat anti-mouse IRDye 680LT secondary antibody (Li-Cor Biosciences, Lincoln, NE) was diluted 1:20,000 into 20 ml of blocking solution containing 0.1% TWEEN-20 and 0.02% SDS and 10 ml was added to the membrane which was then incubated and washed as above except that the membrane was kept in the dark during

incubation and several quick washes in 1× PBS were performed to remove residual milk prior to imaging. Data was acquired using an Odyssey Classic infra-red imager (Li-Cor Biosciences). For the blot shown in *Figure 4—figure supplement 1B*, total cellular protein was electrophoresed and transferred as above except that nitrocellulose membrane (Protran NC, GE Healthcare) was used. Detection of protein was performed using a 1:20,000 dilution of Goat anti-Mouse HRP conjugated secondary antibody, ECL reagents (SuperSignal West, Pierce) and a ChemiDoc XRS + instrument (Bio-Rad). Quantification was performed using ImageLab software.

## Proteins

His-tagged $\sigma^{70}$ and $\sigma^{28}$ were purified from BL21(DE3) cells transformed with pLHN12-His and pET15b-His-fliA, respectively, as described previously (*Panaghie et al., 2000*). *E. coli* core RNAP was purchased from Epicentre (Madison, WI). Holoenzymes were formed by mixing core RNAP and a twofold molar excess of $\sigma^{70}$ or a fivefold molar excess of $\sigma^{28}$ in transcription buffer (20 mM Tris–HCl pH 8.0, 0.1 mM EDTA, 100 mM K-acetate, 100 µg/ml BSA, 10 mM DTT, 5% glycerol, and 0.025% Tween-20) and incubating at 37°C for 10 min.

## In vitro transcription assays

Linear transcription templates were synthesized by PCR using plasmid DNAs (pFW11tet-Ptar_+19, pFW11tet-Ptar_mut+19 or pFW11tet-mutPR′_+19) as template and oligonucleotides that anneal to sequences ~100 bp upstream of the +1 transcription start site (5′ CCTATAAAAATAGGCGTATCAC GAG 3′) and ~135 bp downstream of the transcription termination site ( 5′ CAGGGTTTTCCCAGT CACGACGTTG 3′). 20 nM PCR template was mixed with 15 nM of the appropriate RNAP holoenzyme or RNAP core enzyme in transcription buffer containing 200 µM ATP, 200 µM GTP, 200 µM CTP, 25 µM UTP (supplemented with 0.5 µCi/µL [α-$^{32}$P]-UTP; Perkin Elmer), and 0.5 units/µl Murine RNase Inhibitor (NEB) for 5 min at 37°C to form open complexes. A single round of transcription was initiated by adding MgCl$_2$ (4 mM final concentration) and rifampicin (10 µg/ml final concentration), as described previously (*Grayhack et al., 1985*; *Shankar et al., 2007*; *Hollands et al., 2012*). When present, excess $\sigma^{70}$ was added together with the MgCl$_2$ and rifampicin to a final concentration of 1 µM. Aliquots of the reaction were removed at 1, 6, and 18 min and mixed with 1.2 × stop buffer (600 mM Tris–HCl pH 8.0, 12 mM EDTA, and 100 µg/mL Ambion Yeast RNA [Life Technologies]). Samples were extracted with acid phenol:chloroform and RNA transcripts were recovered by ethanol precipitation and resuspended in gel loading buffer (95% formamide,18 mM EDTA, 0.025% SDS, 0.025% xylene cyanol, 0.025% bromophenol blue, 0.025% amaranth). Samples were heated at 95°C for 5 min, cooled to room temperature, and run on 12% TBE-Urea polyacrylamide gels (UreaGel system; National Diagnostics, Atlanta, GA). Autoradiography of gels was performed using storage phosphor screens and a Typhoon 9400 variable mode imager (GE Healthcare) and quantified using ImageQuant software.

## Acknowledgements

We thank Laura McPartland and MacKenzie Brigham for construction of plasmids, and Jeff Roberts for critical comments on the manuscript.

## Additional information

### Funding

| Funder | Grant reference | Author |
|--------|-----------------|--------|
| National Institute of General Medical Sciences (NIGMS) | GM044025 | Ann Hochschild |
| National Institute of General Medical Sciences (NIGMS) | GM088343 | Bryce E Nickels |

The funder had no role in study design, data collection and interpretation, or the decision to submit the work for publication.

## Author contributions

SRG, NUN, CDW, Conception and design, Acquisition of data, Analysis and interpretation of data; BEN, AH, Conception and design, Analysis and interpretation of data, Drafting or revising the article

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
