## [Decision Letter]

[Editors’ note: a previous version of this study was rejected after peer review, but the authors submitted for reconsideration. The first decision letter after peer review is shown below.]

Thank you for choosing to send your work entitled “The primary σ factor in *Escherichia coli* can access the transcription elongation complex from solution in vivo” for consideration at *eLife*. Your full submission has been evaluated by Richard Losick (Senior Editor), a member of the Board of Reviewing Editors, and three peer reviewers, one of whom, Lucia Rothman-Denes, has agreed to reveal her identity.

We regret to inform you that your work will not be considered further for publication in *eLife*. The decision was reached after discussions between the reviewers.

As you can see from the individual reviews, two of the reviewers were reasonably enthusiastic about your study but one felt the paper fell short of the stated aims. All three reviewers were concerned about the relevance of the σ^70^ overproduction. If this critique can be addressed by additional experiments that identify physiological conditions where σ^70^ levels increase dramatically or examine the effects of an alternative σ factor on transcription elongation initiated from a σ^70^-dependent promoter you might consider sending the manuscript back to *eLife*.

Reviewer #1:

This brief paper is a “proof-of-concept” study that establishes the potential for loading of σ^70^ in *trans* at sites downstream of transcription start sites to regulate transcript elongation in vivo. The authors demonstrate that σ^70^ loads in *trans* at a −10-like element downstream of the λP_R_ promoter in vivo, that it increases pausing at this site, that σ^70^ can load downstream even when the transcript is initiated at a promoter directed by an alternative holoenzyme (Es28; thus establishing that σ^70^ really is loading in trans), that the level of the resulting pause increases in a σ^70^ concentration-dependent manner (further evidence that the effect is exerted in trans), and that the total fraction of transcripts that pause is additive with the effect in *cis* of a σ binding site in the initial transcribed region. The effects are small at natural concentrations of σ^70^ but quite strong when σ^70^ is overproduced. The experiments are well-designed, the results are clear, and the description is precise. These are important findings, building on a proposal from the Landick lab about 10 years ago from experiments with a σ factor fused to core RNAP. The potential that specific operons are regulated by either free σ^70^ or other σ factors at entry sites within the coding regions of operons is enticing. A few ideas for improvements in presentation are listed below, but otherwise the paper is ready to go.

1) It is not clear to me why Figure 5 is presented after the cartoon/summary slide in the Discussion. It seems like it would make more sense to present Figure 5 as Figure 4, just after Figure 3, in the Results section, before the summary slide in the Discussion section. The experiment pictured in Figure 5 is, after all, an important Result. Its presentation in the Discussion makes it seem like an afterthought.

2) Although the technical quality of the paper is high, the importance of σ binding in *trans* to transcription in native operons at levels of σ that exist in nature awaits further studies. The amount of pausing induced by native levels of σ^70^ appears to be fairly low, and there is no nutritional/environmental condition to my knowledge when σ^70^ is overproduced. This should be acknowledged in a more forthright manner, but some potential conditions when such a phenomenon might influence transcription might be expanded on. For example, most of the alternative σ factors are induced naturally, some to quite high levels, and one could imagine that binding sites for these σs at strategic positions in the transcribed region might transiently influence elongation in *trans*.

3) Another direction for the future that might be introduced in the Discussion to add to the impact of the present study would be a genome-wide analysis of the transcriptome following induction of one or more σs. These might turn up candidates for further exploration of in *trans* effects of σ binding at natural levels of σ.

Reviewer #2:

The Hochschild and Nickels labs have previously shown that the initial-transcribed sequences, sequences that lead to pausing of RNA polymerase due to interactions of σ^70^ region 2 with −10 -like sequences, affect the content of σ^70^ in transcription elongation complexes (TECs) in vivo. In addition, they showed that increases in the concentration of σ^70^ increased the efficiency of pausing in vitro.

In this manuscript, they test the in vivo effect of changes in the intracellular concentration of σ^70^ on the proportion of TECs that contain σ^70^. The results presented indicate that in vivo increases in the concentration of σ^70^ lead to increases of its content in TECs whether the TECs originate from σ^70^ or σ^28^-dependent promoters. The results provide further evidence to the growing appreciation of the role of σ factors as a component of the elongation machinery with its implications such as: 1) the ability to affect directly the behavior of the elongation complex through pausing with possible consequences for regulating translation, 2) the ability to compete with or recruit other elongation factors, or 3) trans-regulation of elongation complexes originating from alternative σ factor-dependent promoters.

Requiring clarification:

The experiment in Figure 1 is performed at a 6–7 fold increase in the concentration of σ^70^ over wild-type conditions. Under what experimental conditions is that increase expected in the absence of over-production? Or how sensitive is the response to increases in the concentration of σ^70^.

Reviewer #3:

The manuscript by Goldman et al. aims at demonstrating the action of the *E. coli* primary transcription initiation σ factor, σ^70^, upon transcript elongation, both in vitro and in vivo. Unfortunately, the manuscript under review falls far short of its stated aim and cannot be recommended for publication in *eLife* in its current form.

It is a generally accepted view in the transcription field that shortly after synthesis of 30–50 nt-long RNA by a bacterial RNA polymerase, the initiation/σ factor is released from the transcription complex, likely in competition with nascent RNA and/or elongation factors, such as NusG. Apart from the promoter-proximal σ^70^-dependent pausing, no convincing data exists pointing to the larger role of initiation factor during the elongation phase of transcription. Goldman et al. do not go beyond this already established fact (by focusing on promoter-proximal pausing), nor do they add any new information or nuance to the subject matter. In fact the experiments reported in the manuscript are fairly limited in scope whereas their interpretations are rather over-reaching.

All the experiments reported by Goldman et al. focus on promoter-proximal transcription events, in fact the entire length of the templates used both in vitro and in vivo is ∼116 bp (from the start site to terminator). These offer no new insights into the σ^70^-dependent pausing per se, while the trans-loading onto early elongation complexes formed by σ28-containing holoenzymes is a fairly trivial observation. The significance of the manuscript's “findings” is dramatically overstated: e.g. the “control” of elongation by σ^70^ in vitro is not quantified in terms of elongation rate, pausing efficiency or pause half-life, instead the reader is presented with a (phosphor)image (Figure 2) s/he can only eyeball. Based on such eyeballing this reviewer concludes that the σ^70^ effects on elongation are minimal (judging by the apparent “amount” of full-length product) and limited to the induction of promoter-proximal arrest or termination in a small fraction of complexes. These complexes are labeled as “pause” on Figure 2 without any merit: there is no apparent time-decay (escape from the pause site), and no discernable escape from these “pauses” for up to 18 min (lanes 6 and 18). Moreover, it was not determined whether or not the shorter transcripts were still associated with RNA polymerase (which would distinguish paused/arrested complexes from terminated ones). As such, all that can be asserted from these experiments is the changes in RNA length. The same applies to the in vivo experiments reported by Goldman et al. Physical loading of σ^70^ onto elongation complexes has not been demonstrated either in vivo or in vitro which in turn means that Goldman et al. should limit their conclusions to a simple correlation between the concentration of σ^70^ and the abundance of the short transcripts produced from a handful of artificial templates.

The technical aspect of the work reported by Goldman et al. is confusing. This is especially evident in the description of their in vitro experiments: the authors claim to initiate transcription in vitro by adding together MgCl2 (which supplies catalytic Mg^2+^) and rifampicin (which inhibits transcription initiation by restricting nascent RNA synthesis to the first 3–4 nt) to the pre-formed open complexes. In single-round transcription assays, rifampicin is routinely added after the elongation complex is stalled (by selective NTP deprivation) in proximity to the promoter (and is no longer sensitive to it) to prevent re-initiation. As described by Goldman et al. the single-round assay would be a no-round one and the transcripts depicted in Figure 2 could not have been obtained as described in the Materials and methods.

Furthermore, in order to make such assays quantitative the radioactively labeled nucleotide is usually supplied at the initiation stage and is incorporated into a defined length of RNA (essentially an end-labeling). This nucleotide is later removed (e.g. by gel-filtration) or diluted to negligible levels by the excess of the unlabeled one in the chase/extension mix. This way the amount of radioactivity in RNA is practically length-independent, enabling quantitative kinetic analysis of pausing, termination, etc. In contrast, in experiments described by Goldman et al. both labeled and un-labeled UTP are present in the reaction mix together from the start and no measure is mentioned that was taken to remove/dilute P32-labeled one. This makes the amount of radioactivity in the transcript dependent on its length and sequence (more radioactivity is incorporated as the transcript grows) rendering any quantitative interpretation of the data impractical.

The conclusions Goldman et al. derive from their in vivo experiments call for greater scope and technical sophistication of experimentation than the ones employed (i.e. monitoring the levels of transcripts produced from 2 short artificial templates by LNA hybridization and of 2 proteins by Western blotting). High-throughput/deep sequencing and mass-spectrometry allow for much more rigorous and quantitative monitoring of the proposed control σ^70^ exerts over various stages of transcription in vivo (and incidentally of the impact its overexpression would have on the proteome outside its own concentration). These methods are presently accessible and affordable enough to make the approach chosen by Goldman et al. seem somewhat myopic in scope, and archaic in design.

[Editors’ note: what now follows is the decision letter after the authors submitted for further consideration.]

Thank you for submitting your work entitled “The primary σ factor in *Escherichia coli* can access the transcription elongation complex from solution in vivo” for peer review at *eLife*. Your submission has been favorably evaluated by Richard Losick (Senior Editor), a Reviewing Editor, and three reviewers, one of whom, Lucia Rothman-Denes, has agreed to reveal her identity.

The reviewers discussed the reviews with one another and all felt the manuscript has significantly improved over the first submission. The Reviewing Editor has drafted this decision to help you prepare a revised submission.

Previous in vitro experiments showed that σ^70^ increases the efficiency of pausing at −10-like elements when supplied from solution (trans-loading). This study demonstrates that trans-loading occurs in vivo; σ^70^ can associate with a transcription complex initiated from an upstream σ^28^-dependent promoter and affect pausing of the complex, particularly in stationary phase.

Essential revisions:

1) The reviewers were not able to reach a consensus on whether genome-wide ChIP-seq and transcriptome-wide RNASeq studies were needed to address the physiological significance of trans-loading. The authors should discuss whether trans-loading is just an accidental process or plays a meaningful role in the cell and what approaches will be needed to address the biological significance of trans-loading in the future.

2) In experiments designed to compare the level of transcripts originating from plasmids in exponentially growing cells with those in stationary phase no effort was made to ascertain whether the plasmid levels (copy numbers) remained equal at the two growth stages (most likely were not). The authors should determine copy number under the two conditions or explain why this is not a valid concern.

3) Figure 2: Sequences upstream of +1 should be added up to the −35 element, since these determine the appearance of the (*) transcript.

4) Figure 4. Figure heading needs to be corrected: top lane is template; second lane is growth phase.

---

## [Author Response]

[Editors’ note: the author responses to the first round of peer review follow.]

We thank the reviewers for their comments, which we respond to in detail below. We believe that our revised manuscript addresses the main overall critique of our original manuscript – namely that we had not identified a physiological condition under which σ^70^
*trans* loading could be expected to have a significant effect on transcription in the cell. We now show that the effect of σ^70^
*trans* loading on transcription pausing becomes highly significant in the absence of ectopically produced σ^70^ when the cells are in stationary phase. In particular, we observe an −10-fold increase in the abundance of the pause species in stationary-phase cells as compared to exponential-phase cells (see Figure 4 of the revised manuscript). The use of a σ^28^-dependent transcription unit to demonstrate this growth phase- dependent effect of σ^70^ on transcription pausing establishes unequivocally that the effect is due to *trans*-loaded σ^70^ and not σ^70^ that has been retained following the transition from initiation to elongation. Additionally, in response to reviewer #3, we have shown that the short transcripts that we identify as pause RNAs are in fact RNAP-associated, confirming their identify as pause species.

*[…] As you can see from the individual reviews, two of the reviewers were reasonably enthusiastic about your study but one felt the paper fell short of the stated aims. All three reviewers were concerned about the relevance of the σ*^*70*^
*overproduction. If this critique can be addressed by additional experiments that identify physiological conditions where σ*^*70*^
*levels increase dramatically or examine the effects of an alternative σ factor on transcription elongation initiated from a σ*^*70*^*-dependent promoter you might consider sending the manuscript back to* eLife*.*

Reviewer #1:

*1) It is not clear to me why*
Figure 5
*is presented after the cartoon/summary slide in the Discussion. It seems like it would make more sense to present*
Figure 5
*as*
Figure 4*, just after*
Figure 3*, in the Results section, before the summary slide in the Discussion section. The experiment pictured in*
Figure 5
*is, after all, an important Result. Its presentation in the Discussion makes it seem like an afterthought*.

We agree that this Figure was awkwardly placed. We now present this result in Figure 1 (as panel B), after showing the effect of σ^70^ overproduction on the template that bears the single transcribed-region pause element.

*2) Although the technical quality of the paper is high, the importance of σ binding in* trans *to transcription in native operons at levels of σ that exist in nature awaits further studies. The amount of pausing induced by native levels of σ*^*70*^
*appears to be fairly low, and there is no nutritional/environmental condition to my knowledge when σ*^*70*^
*is overproduced. This should be acknowledged in a more forthright manner, but some potential conditions when such a phenomenon might influence transcription might be expanded on. For example, most of the alternative σ factors are induced naturally, some to quite high levels, and one could imagine that binding sites for these σs at strategic positions in the transcribed region might transiently influence elongation in trans*.

In our revised manuscript, we describe a condition that leads to a high degree of pausing in the absence of ectopically produced σ^70^. In particular, we now show that the effect of σ^70^
*trans* loading on transcription pausing manifests a dramatic growth phase dependence. Thus, we observe an ∼10-fold increase in the abundance of the pause species in stationary-phase cells as compared to exponential-phase cells (see Figure 4 of the revised manuscript). The amount of pausing we detect in stationary phase cells containing chromosomally encoded σ^70^ only is greater than that detected during exponential phase with σ^70^ overproduction. A rifampicin time-course experiment suggests that the observed increase in the abundance of the pause species is due, at least in part, to a decrease in pause half-life (Figure 4). Although we did not detect an increase in the total cellular concentration of σ^70^ in stationary phase, we cannot exclude the possibility that the amount of free σ^70^ available to bind the TEC varies with growth phase.

*3) Another direction for the future that might be introduced in the Discussion to add to the impact of the present study would be a genome-wide analysis of the transcriptome following induction of one or more σs. These might turn up candidates for further exploration of in* trans *effects of σ binding at natural levels of σ*.

We agree with the reviewer and intend to use NET-seq (with an epitope-tagged version of σ^70^) to perform such genome-wide analyses.

Reviewer #2:

*Requiring clarification*:

*The experiment in*
Figure 1
*is performed at a 6–7 fold increase in the concentration of σ*^*70*^
*over wild-type conditions. Under what experimental conditions is that increase expected in the absence of over-production? Or how sensitive is the response to increases in the concentration of σ*^*70*^.

See response to reviewer #1, point 2, describing our new data. With regard to the sensitivity of the response, we note that in the experiment of Figure 3 (with the P*tar* template) a comparable increase in σ^70^-dependent pausing is observed with only an ∼3-fold increase in the concentration of σ^70^.

Reviewer #3:

*[…] It is a generally accepted view in the transcription field that shortly after synthesis of 30-50 nt-long RNA by a bacterial RNA polymerase, the initiation/σ factor is released from the transcription complex, likely in competition with nascent RNA and/or elongation factors, such as NusG. Apart from the promoter-proximal σ*^*70*^*-dependent pausing, no convincing data exists pointing to the larger role of initiation factor during the elongation phase of transcription. Goldman et al. do not go beyond this already established fact (by focusing on promoter-proximal pausing), nor do they add any new information or nuance to the subject matter. In fact the experiments reported in the manuscript are fairly limited in scope whereas their interpretations are rather over-reaching*.

Although σ^70^-dependent pausing has indeed been well established, our finding that such pausing can occur in vivo due to *trans* loaded σ^70^ is new. We believe that our revised manuscript provides strong evidence for physiological relevance with the new demonstration that the abundance of paused species due to *trans* loaded σ^70^ is dramatically increased during stationary phase.

*All the experiments reported by Goldman et al. focus on promoter-proximal transcription events, in fact the entire length of the templates used both in vitro and in vivo is ∼ 116 bp (from the start site to terminator). These offer no new insights into the σ*^*70*^*-dependent pausing per se, while the trans-loading onto early elongation complexes formed by σ*^*28*^*-containing holoenzymes is a fairly trivial observation*.

Our experimental system enables the detection of pausing events that occur relatively close to the promoter. But as these events are unambiguously occurring during the elongation phase of transcription, we do not believe their significance is diminished by their promoter-proximal location. We note that the 5’- untranslated region is the locus of many well characterized examples of post-initiation transcription regulation, and that early elongation pausing is emerging as a theme in both prokaryotic and eukaryotic transcription. With also wish to emphasize that the use of a σ^28^-dependent transcription unit for our experiments enables us to distinguish unequivocally between pausing due to retained σ^70^ and pausing due to *trans* loaded σ^70^. And, as noted above, the growth phase dependence of pausing due to *trans* loaded σ^70^ that we now report suggests that such pausing has the potential to influence the entire transcriptome.

*The significance of the manuscript's “findings” is dramatically overstated: e.g. the “control” of elongation by σ*^*70*^
*in vitro is not quantified in terms of elongation rate, pausing efficiency or pause half-life, instead the reader is presented with a (phosphor)image (*Figure 2*) s/he can only eyeball. Based on such eyeballing this reviewer concludes that the σ*^*70*^
*effects on elongation are minimal (judging by the apparent “amount” of full-length product) and limited to the induction of promoter-proximal arrest or termination in a small fraction of complexes. These complexes are labeled as “pause” on*
Figure 2
*without any merit: there is no apparent time-decay (escape from the pause site), and no discernable escape from these “pauses” for up to 18 min (lanes 6 and 18)*.

We wish to clarify that the purpose of our study is to investigate whether or not σ^70^
*trans* loading occurs in vivo, and what it’s potential impact might be. As cited in the manuscript, σ^70^
*trans* loading in vitro has been demonstrated previously. The point of the in vitro transcription experiments shown in Figure 2 is simply to establish a correspondence between the pause species that arise when transcription initiates from ^λP^_R'_ (as reported previously) and those that arise when transcription initiates from a σ^28^-dependent promoter. Our previous study with the ^λP^_R'_ template (8) established the identity of the ∼35-nt RNAs as σ^70^-dependent pause species based on 4 characteristics, including their sensitivity to the transcript cleavage factor GreB and their sensitivity to a mutation in σ^70^ that causes a defect in σ^70^-dependent pausing. In the context of the current study, we did not think it appropriate to repeat these tests. In the revised manuscript, we now clarify that the transcription reactions were performed in the absence of GreA and GreB, which accounts for the long-lived pause species (see legend to Figure 2); in Deighan et al., we show that the pause species chase very quickly when GreB is added to the reactions.

*Moreover, it was not determined whether or not the shorter transcripts were still associated with RNA polymerase (which would distinguish paused/arrested complexes from terminated ones). As such, all that can be asserted from these experiments is the changes in RNA length. The same applies to the in vivo experiments reported by Goldman et al. Physical loading of σ*^*70*^
*onto elongation complexes has not been demonstrated either in vivo or in vitro which in turn means that Goldman et al. should limit their conclusions to a simple correlation between the concentration of σ*^*70*^
*and the abundance of the short transcripts produced from a handful of artificial templates*.

In our revised manuscript, we have addressed the reviewer’s concern about whether or not the short (∼35-nt) RNA species observed in vivo actually represent pause RNAs. Specifically, we have shown that these species are RNAP-associated by performing immunoprecipitation experiments using a strain that encodes the RNAP β’ subunit with a C-terminal 3xFLAG tag (see new panel B of Figure 3 and Figure 4—figure supplement 1).

*The technical aspect of the work reported by Goldman et al. is confusing. This is especially evident in the description of their in vitro experiments: the authors claim to initiate transcription in vitro by adding together MgCl2 (which supplies catalytic Mg*^*2+*^*) and rifampicin (which inhibits transcription initiation by restricting nascent RNA synthesis to the first 3–4 nt) to the pre-formed open complexes. In single-round transcription assays, rifampicin is routinely added after the elongation complex is stalled (by selective NTP deprivation) in proximity to the promoter (and is no longer sensitive to it) to prevent re-initiation. As described by Goldman et al. the single-round assay would be a no-round one and the transcripts depicted in*
Figure 2
*could not have been obtained as described in the Materials and methods*.

We neglected to specify in the original manuscript that this is a well-established procedure for initiating single-round transcription, which is effective because most of the transcription complexes escape into productive elongation before rifampicin reaches its binding site on RNAP. We now clarify this point by citing several prior studies that use this method to initiate single-round transcription (including two of a large number of papers from the Roberts lab over a 30-year period).

*Furthermore, in order to make such assays quantitative the radioactively labeled nucleotide is usually supplied at the initiation stage and is incorporated into a defined length of RNA (essentially an end-labeling). This nucleotide is later removed (e.g. by gel-filtration) or diluted to negligible levels by the excess of the unlabeled one in the chase/extension mix. This way the amount of radioactivity in RNA is practically length-independent, enabling quantitative kinetic analysis of pausing, termination, etc. In contrast, in experiments described by Goldman et al. both labeled and un-labeled UTP are present in the reaction mix together from the start and no measure is mentioned that was taken to remove/dilute P32-labeled one. This makes the amount of radioactivity in the transcript dependent on its length and sequence (more radioactivity is incorporated as the transcript grows) rendering any quantitative interpretation of the data impractical*.

As we hope we have clarified, we are not attempting to make any quantitative argument about the in vitro data. The focus of our study is the question of what occurs in vivo.

*The conclusions Goldman et al. derive from their in vivo experiments call for greater scope and technical sophistication of experimentation than the ones employed (i.e. monitoring the levels of transcripts produced from 2 short artificial templates by LNA hybridization and of 2 proteins by Western blotting). High-throughput/deep sequencing and mass-spectrometry allow for much more rigorous and quantitative monitoring of the proposed control σ*^*70*^
*exerts over various stages of transcription in vivo (and incidentally of the impact its overexpression would have on the proteome outside its own concentration). These methods are presently accessible and affordable enough to make the approach chosen by Goldman et al. seem somewhat myopic in scope, and archaic in design*.

We believe that a genome-wide analysis of the effects of σ^70^
*trans* loading is beyond the scope of the current study, though we certainly agree that such experiments will be revealing. We do not agree with the reviewer’s assessment of our approaches in the study, which we believe are well suited to address the questions we sought to investigate.

[Editors' note: the author responses to the re-review follow.]

*Essential revisions*:

*1) The reviewers were not able to reach a consensus on whether genome-wide ChIP-seq and transcriptome-wide RNASeq studies were needed to address the physiological significance of trans-loading. The authors should discuss whether trans-loading is just an accidental process or plays a meaningful role in the cell and what approaches will be needed to address the biological significance of trans-loading in the future*.

We have added a sentence to the last paragraph of the Discussion in which we specify what approaches will be useful in the future to identify transcription units that manifest pausing that is attributable to *trans*-loaded σ^70^.

*2) In experiments designed to compare the level of transcripts originating from plasmids in exponentially growing cells with those in stationary phase no effort was made to ascertain whether the plasmid levels (copy numbers) remained equal at the two growth stages (most likely were not). The authors should determine copy number under the two conditions or explain why this is not a valid concern*.

We agree that plasmid copy number might differ between exponentially growing cells and stationary phase cells. However, we do not believe any such difference would invalidate our conclusion that the extent of pausing due to *trans*-loaded σ^70^ varies with growth phase. Specifically, we show that pause half-life increases significantly during stationary phase, an effect that could not be explained by changes in plasmid copy number. We conclude in the manuscript that the observed effect of growth phase on the extent of pausing is due “at least in part” to an increase in the half-life of the pause (Results, last sentence and Discussion, subsection “Dual pathways for σ^70^ to associate with the TEC in vivo”, second paragraph).

*3)*
Figure 2*: Sequences upstream of +1 should be added up to the -35 element, since these determine the appearance of the (*) transcript*.

We have included the sequences upstream of +1 for each promoter in the Materials and methods (and cited this in the legend to Figure 2). We note that the appearance of the (*) transcript can be explained by the fact that the pause element is an extended −10 element (TGNTATAAT).

*4)*
Figure 4*. Figure heading needs to be corrected: top lane is template; second lane is growth phase*.

The figure heading has been corrected.